# Novel GPR18 Ligands in Rodent Pharmacological Tests: Effects on Mood, Pain, and Eating Disorders

**DOI:** 10.3390/ijms24109046

**Published:** 2023-05-20

**Authors:** Małgorzata Frankowska, Karolina Wydra, Agata Suder, Magdalena Zaniewska, Dawid Gawliński, Joanna Miszkiel, Anna Furgała-Wojas, Kinga Sałat, Małgorzata Filip, Christa E. Müller, Katarzyna Kieć-Kononowicz, Magdalena Kotańska

**Affiliations:** 1Department of Drug Addiction Pharmacology, Maj Institute of Pharmacology Polish Academy of Sciences, Smętna 12 Street, 31-343 Kraków, Poland; 2Chair of Pharmacodynamics, Faculty of Pharmacy, Jagiellonian University Medical College, Medyczna 9 Street, 30-688 Kraków, Poland; 3PharmaCenter Bonn, Pharmaceutical Institute, Pharmaceutical & Medicinal Chemistry, University of Bonn, An der Immenburg 4, D-53121 Bonn, Germany; 4Chair of Technology and Biotechnology of Drugs, Faculty of Pharmacy, Jagiellonian University Medical College, Medyczna 9 Street, 30-688 Kraków, Poland; 5Department of Pharmacological Screening, Faculty of Pharmacy, Jagiellonian University Medical College, Medyczna 9 Street, 30-688 Kraków, Poland

**Keywords:** anxiety, depression, food intake, GPR18 ligands, locomotor activity, pain, rodents, temperature, Δ^9^-THC, Δ^9^-THC-vehicle discrimination

## Abstract

The lack of selective pharmacological tools has limited the full unraveling of G protein-coupled receptor 18 (GPR18) functions. The present study was aimed at discovering the activities of three novel preferential or selective GPR18 ligands, one agonist (PSB-KK-1415) and two antagonists (PSB-CB-5 and PSB-CB-27). We investigated these ligands in several screening tests, considering the relationship between GPR18 and the cannabinoid (CB) receptor system, and the control of endoCB signaling over emotions, food intake, pain sensation, and thermoregulation. We also assessed whether the novel compounds could modulate the subjective effects evoked by Δ^9^-tetrahydrocannabinol (THC). Male mice or rats were pretreated with the GPR18 ligands, and locomotor activity, depression- and anxiety-like symptoms, pain threshold, core temperature, food intake, and THC-vehicle discrimination were measured. Our screening analyses indicated that GPR18 activation partly results in effects that are similar to those of CB receptor activation, considering the impact on emotional behavior, food intake, and pain activity. Thus, the orphan GPR18 may provide a novel therapeutic target for mood, pain, and/or eating disorders, and further investigation is warranted to better discern its function.

## 1. Introduction

G-protein-coupled receptors (GPCRs) are one of the largest and most diverse, physiologically important membrane proteins that are crucial for physiological processes, as they sense signaling molecules, such as hormones and neurotransmitters. They serve as the communication interface between the cell’s external and internal environments, and may be targeted by new drugs for therapeutic purposes. GPCRs contain a seven-transmembrane helical bundle that provides a binding site for their ligands. The ligand binding triggers a slight change in GPCR conformation that is propagated through the whole protein, ultimately causing alterations at the receptor’s cytoplasmic surface that permit binding to its cognate G protein, followed by activation of complex cytosolic signaling networks, resulting in a cellular response [1,2].

The G protein-coupled receptor 18 (GPR18), first described by Gantz and coworkers in 1997, was initially described to act through Gαi/o, Gαq/11, and Gαs proteins [3]; however, this signaling has not been unambiguously confirmed by us and others [4,5]. In humans, GPR18 mRNA transcripts were reported to be expressed in gastrointestinal, immune, and testicular tissues, sperm, and metastatic melanoma, as well as various brain structures (mainly hypothalamus, cerebellum, and brain stem) [6,7,8]. Based on the above localization, several lines of evidence further demonstrated GPR18 engagement in pain sensation, sperm physiology, immunomodulation, intraocular pressure, metabolism, and cancer [3].

Despite sharing low sequence homology to cannabinoid (CB)_1_ and CB_2_ receptors (13% and 8%, respectively), and to the putative CB receptor GPR55 (14–20%) [3,9], GPR18 has been proposed as a new CB receptor candidate since endogenous endocannabinoid metabolites (e.g., *N*-arachidonoylglycine (NAGly) [6,7,10,11], phytogenic cannabinoids (e.g., Δ^9^-tetrahydrocannabinol (THC)), and synthetic cannabinoids (e.g., *N*-arachidonoylcyclopropylamide, O-1602) [9] were claimed to activate GPR18. Among these, only THC has been confirmed to be a moderately potent GPR18 agonist up to now, but it is much more potent at CB receptors [5,6].

The endogenous polyunsaturated fatty acid metabolite resolvin D29 (RvD2) was also reported to stimulate GPR18 [12], but again, this has not been independently confirmed, and GPR18 still remains an orphan receptor of the class A, rhodopsin-like receptor family [10]. Of note, GPR18 may also heterodimerize with CB receptors [13].

The lack of selective pharmacological tools for many years has limited the full unraveling of GPR18’s physiological and pathological functions. This study aimed to uncover the biological activities of a new GPR18 agonist (PSB-KK-1415: hGPR18 − EC_50_ = 0.0191 μM, β-arrestin recruitment assay) and two GPR18 antagonists (PSB-CB-5: IC_50_ = 0.279 μM, and PSB-CB-27: IC_50_ = 0.650 μM, inhibition of THC activation of GPR18 in β-arrestin recruitment assay) that show GPR18 selectivity. These are unique tools to assess the biological activities of GPR18 [14,15]. Since all previously used antagonists of the GPR18 also antagonize the GPR55 [16,17], it has been difficult to separate the effects mediated by these two receptor targets. Hence, the discovery of selective GPR18 ligands is useful for research focused on the assessment of the biological role of this receptor.

Considering the structural relationship between the GPR18 and CB receptor system, as well as endoCB signaling control over emotions, food intake, pain sensation, and thermoregulation, we employed several screening tests assessing locomotor activity, depression-like and anxiety-like symptoms, pain threshold, core temperature, and food intake to study the effects of novel GPR18 ligands. We additionally assessed whether these compounds could influence THC-vehicle discrimination. The latter procedure—used extensively as a preclinical assay in drug development—helps us to classify drugs based on shared discriminative stimulus properties, and the results reflect their specific CNS activity at neurotransmitter receptors [18].

Our screening analyses indicated that stimulation of GPR18 partly results in effects that are similar to those of CB receptor activation, considering the impact on emotional behavior, food intake, and pain perception.

## 2. Results

### 2.1. Locomotor Activity

Acute administration of PSB-KK-1415 did not change the locomotor activity in drug-naïve mice either after 60 or 120 min (H = 3.00, *p* = 0.56 and H = 3.08, *p* = 0.55, respectively) (Figure 1). Following acute PSB-CB-5 administration, the total locomotor activity was significantly reduced after 60 and 120 min (H = 18.09, *p* < 0.001 and H = 13.44, *p* < 0.01, respectively; Figure 1). The post hoc Dunn’s test showed a significant reduction of animal activity after PSB-CB-5 at 3 and 30 mg/kg, but not at 1 and 10 mg/kg, compared to vehicle-treated animals after 60 and 120 min (*p* < 0.001 and *p* < 0.05, respectively). Similarly, a reduction in locomotor activity of the mice was observed for PSB-CB-27 after 60 and 120 min (H = 12.15, *p* < 0.05 and H = 18.95, *p* < 0.01, respectively). PSB-CB-27 at 10, but not 0.1–3 mg/kg, significantly (*p* < 0.05) reduced locomotor activity recorded after 60 and 120 min following its administration (Figure 1).

### 2.2. Forced Swim Test

Acute administration of PSB-KK-1415 significantly changed the immobility time in naïve mice (H = 15.03, *p* < 0.01; Figure 2a). The post hoc Dunn’s test revealed that PSB-KK-1415 at a dose 30 mg/kg attenuated immobility time compared to the control group (*p* < 0.05). Neither PSB-CB-5 nor PSB-CB-27 showed significant effects on the immobility time (H = 1.76, *p* = 0.62 and H = 4.48, *p* = 0.34, respectively; Figure 2a).

### 2.3. Four-Plate Test

PSB-KK-1415 significantly increased the number of spontaneous punished crossings (H = 13.29, *p* < 0.01; Figure 2b), and a significant effect was observed only for a dose of 1 mg/kg (*p* < 0.05). PSB-CB-5 significantly altered the number of punished crossings (H = 8.33, *p* < 0.05). PSB-CB-5 at a dose of 10, but not 1 and 3 mg/kg, significantly increased the number of spontaneous punished crossings (Figure 2b). PSB-CB-27 was inactive in this test; none of the PSB-CB-27 doses affected the number of punished crossings (H = 5.91, *p* = 0.21; Figure 2b).

### 2.4. Food Intake

PSB-KK-1415 attenuated food intake in mice during a 2-h observation period (H = 13, *p* = 0.011; Figure 2c). The post hoc Dunn’s test showed a significant (*p* < 0.01) reduction in food consumption by mice after PSB-KK-1415 given at a dose of 30 mg/kg, but not at 1–10 mg/kg, compared to vehicle-treated animals (Figure 2c). PSB-CB-5 (1–10 mg/kg) attenuated food intake in mice during a 2-h observation period (H = 11.52, *p* = 0.0092; Figure 2c). Post hoc Dunn’s test revealed that PSB-CB-5 at a dose of 3 and 10 mg/kg attenuated food consumption in mice (*p* < 0.01 and *p* < 0.05, respectively; Figure 2c). PSB-CB-27 (0.1–3 mg/kg) did not change food intake in mice during a 2-h observation period (H = 3.25, *p* = 0.52; Figure 2c).

### 2.5. Hot Plate Test (Acute Pain)

Treatment with PSB-KK-1415 at doses of 1–30 mg/kg did not demonstrate analgesic properties in this assay (one-way ANOVA for repeated measures did not reveal drug × time interaction (F(16,112) = 0.80, *p* = 0.68, Figure 3 left). Overall treatment with PSB-CB-5 at doses of 1–10 mg/kg did not demonstrate analgesic properties (ANOVA for repeated measures did not reveal drug × time interaction: F(12,84) = 0.84, *p* = 0.61, Figure 3 middle). For PSB-CB-27, the statistical analysis showed the following ANOVA values: time effect (F(4,28) = 3.95, *p* < 0.05), drug × time interaction (F(16,112) = 2.57, *p* < 0.01), but no drug effect (F(4,28) = 1.11, *p* = 0.37; Figure 3 right). The post hoc Tukey’s analysis showed that PSB-CB-27 at a separate dose of 0.3 mg/kg significantly (at least *p* < 0.01) reduced latency time from 30 to 60 min post-treatment compared to time 0.

### 2.6. Oxaliplatin-Induced Neuropathic Pain (Chronic, Neuropathic Pain Model)

#### 2.6.1. Effect on the Mechanical Nociceptive Threshold in Oxaliplatin-Treated Mice

In the von Frey test, statistical evaluation by two-way repeated measures ANOVA revealed an overall effect of treatment (F(4,45) = 40.70, *p* < 0.0001). Time affected the results significantly (F(4,180) = 146.04, *p* < 0.0001) and the drug × time interaction was also significant (F(16,180) = 7.45, *p* < 0.0001; Figure 4a,b).

A detailed analysis of changes within individual groups performed before and after treatment with oxaliplatin revealed that during the early phase of tactile allodynia (day 1) in all experimental groups, oxaliplatin significantly lowered the pain threshold for mechanical stimulation (*p* < 0.0001 vs. paw withdrawal threshold before oxaliplatin injection). On day 7, tactile allodynia was also noted in all experimental groups (*p* < 0.0001 vs. value before oxaliplatin injection). In this phase, the compound PSB-KK-1415 at the dose of 30 mg/kg elevated the mechanical pain threshold in oxaliplatin-treated mice (significant at *p* < 0.01 vs. pre-drug paw withdrawal force; Figure 4a).

#### 2.6.2. Effect on the Thermal (Cold) Nociceptive Threshold in Oxaliplatin-Treated Mice

Statistical evaluation by two-way repeated measures ANOVA revealed an overall effect of treatment (F(4,40) = 27.81, *p* < 0.0001) in the cold plate test. Time affected the results significantly (F(4,160) = 125.72, *p* < 0.0001) and the drug × time interaction was also significant (F(16,160) = 6.64, *p* < 0.0001).

A detailed analysis of changes within individual groups performed before and after treatment with oxaliplatin revealed that in the early phase of cold hyperalgesia (day 1) in all experimental groups, oxaliplatin significantly lowered the pain threshold for cold stimulation (*p* < 0.0001 vs. latency before oxaliplatin injection; Figure 4c,d). On day 7, cold hyperalgesia was also noted in all experimental groups (*p* < 0.0001 vs. latency before oxaliplatin injection). In this phase of neuropathy, none of the compounds tested was effective in reducing cold hypersensitivity in oxaliplatin-treated mice. Of note, the compound PSB-CB-27 at the dose of 30 mg/kg lowered the cold nociceptive threshold of oxaliplatin-treated mice (significant at *p* < 0.01 vs. pre-drug latency measured on day 7; Figure 4d).

### 2.7. Body Temperature

The groups of mice tested for the effect of PSB-CB-5 (1–30 mg/kg), PSB-CB-27 (0.1–10 mg/kg), THC, or a combination of PSB-CB-5 or PSB-CB-27 and THC did not differ in basal rectal body temperature; the initial body temperature was 37.02–37.29 ± 0.05–0.08 °C (Figure 5a–d, left).

#### 2.7.1. Effects of PSB-CB-5 and PSB-CB-27

An ANOVA for repeated measures indicated an effect for the drug × time interaction (F(24,150) = 2.30, *p* < 0.01), time (F(6,150) = 27.3, *p* < 0.001) and the drug (F(4,25) = 2.20, *p* = 0.10; Figure 5a, right) on rectal body temperature in mice treated with PSB-CB-5 (Figure 5a, right). The post hoc Tukey’s test indicated increased body temperature during the time for all groups treated with PSB-CB-5. PSB-CB-27 (0.1–10 mg/kg) altered mouse rectal body temperature and an ANOVA for repeated measures indicated an effect for the drug × time interaction (F(30,186) = 2.50, *p* < 0.001), for time (F(6,186) = 16.30, *p* < 0.001), but not for the drug (F(5,31) = 1.60, *p* = 0.18; Figure 5b, right). The post hoc Tukey’s analysis showed that PSB-CB-27 at a separate dose of 10 mg/kg significantly (*p* < 0.05) decreased body temperature 30 min after injection compared to the control group.

#### 2.7.2. Combination Studies of PSB-CB-5 or PSB-CB-27 and THC

THC (10–20 mg/kg) dose-dependently reduced body temperature in mice (ANOVA for repeated measures showed an effect for the drug (F(2,17) = 6.56, *p* < 0.01), for time (F(6,102) = 2.27, *p* < 0.05), and for the drug × time interaction (F(12,102) = 2.33, *p* < 0.05; Figure 5c, right). The post hoc Tukey’s analysis indicated a significant (at least *p* < 0.01) reduction in body temperature after THC at a dose of 10 mg/kg seen from 15–90 min of measurement, and at dose 20 mg/kg seen from 0–120 min of measurement.

Co-administration of PSB-CB-5 or PSB-CB-27 with THC did not change the hypothermic effect of THC. Statistical analysis for co-administered THC (20 mg/kg) and PSB-CB-5 (3 mg/kg) indicated a significant effect for the drug (F(1,12) = 26.90, *p* < 0.001) and for the drug × time interaction (F(6,72) = 3.86, *p* < 0.01), but not for time (F(6,72) = 1.26, *p* = 0.29), Figure 5d, right). The post hoc Tukey’s test revealed a reduction in body temperature for all times of measurement (*p* < 0.001) compared to the control group. Similarly, co-administration of THC (20 mg/kg) and PSB-CB-27 (1–3 mg/kg) reduced rectal temperature in mice (ANOVA for repeated measures: effect for the drug F(2,18) = 11.91, *p* < 0.01, time F(6,108) = 6.98 *p* < 0.001, and the drug × time interaction F(12,108) = 4.29, *p* < 0.001; Figure 5d, right). The post hoc Tukey’s test showed a reduction in body temperature from 0–120 min (*p* < 0.01) for both PSB-CB-27 doses given in combination with THC compared to the control group.

### 2.8. Drug Discrimination

#### 2.8.1. THC—Saline Discrimination

THC (3 mg/kg) versus saline discrimination was observed in an average of 32 sessions. Administration of THC (0.3–3 mg/kg) to rats produced a dose-dependent increase in drug lever pressing, whereas saline administration resulted in only 5% of the maximal lever pressing response observed for THC (Figure 6). The drug-lever responses after 0.3 and 1 mg/kg of THC were significantly different from the preceding training session (*p* < 0.05). The dose of THC predicted to elicit 50% of the maximal THC-induced lever pressing response (ED_50_) in rats was 1 mg/kg (95% CL 1.39 ± 1.41 mg/kg). Response rates for all test doses of THC and saline did not differ from those obtained during the immediately preceding THC maintenance session (*p* > 0.05; Figure 6).

#### 2.8.2. Substitution Studies

At the doses tested, PSB-KK-1415, PSB-CB-5, and PSB-CB-27 evoked a maximum of 34% drug-lever responding when given alone, which indicated no substitution for the THC training dose (3 mg/kg). Neither PSB-KK-1415 (1–10 mg/kg) nor PSB-CB-5 (10 mg/kg) and PSB-CB-27 (10 mg/kg) altered the response rates of the animals as compared to previous THC and saline training sessions (Figure 7a and Figure 8a). Additionally, PSB-KK-1415 (10 mg/kg) did not evoke significant results for its treatment for five different time points (from 5 to 90 min; Figure 7a), as assessed by Friedman ANOVA (F(4,20) = 4.25, *p* = 0.37).

#### 2.8.3. Combination Studies

The Kruskal–Wallis H test with multiple comparisons (2-tailed) tests revealed that pretreatment with PSB-KK-1415 (10 mg/kg) plus THC (0.3–3 mg/kg) did not yield different results from those of the THC maintenance session for drug-lever responding (H = 12.72, *p* = 0.03) (Figure 7b). Similarly, using an one-way ANOVA analysis for the response rate, assessed as a number of responses per min (F(5,37) = 4.90, *p* < 0.001), and the post-hoc Uniqal N tests, pretreatment with PSB-KK-1415 (10 mg/kg) plus THC (0.3–3 mg/kg) did not yield different results from those of the previous THC maintenance session (Figure 7b).

Pretreatment with PSB-CB-5 at a dose of 10 mg/kg combined with various doses of THC (0.3–3 mg/kg) altered the THC-lever responding, as assessed by the Kruskal–Wallis test for the drug-lever responding (H = 14.40, *p* < 0.01) and for the response rates (H = 17.63, *p* < 0.01); however, multiple comparison (2-tailed) tests showed that this combination did not alter the THC-lever responding (Figure 8b).

The Kruskal–Wallis H test for pretreatment with PSB-CB-27 at a dose of 10 mg/kg combined with various doses of THC (0.3–3 mg/kg) indicated a significant effect for the drug-lever responses (H = 14.45, *p* < 0.01), but not for the response rates (H = 9.98, *p* = 0.08). Further multiple comparisons (2-tailed) tests revealed that pretreatment with PSB-CB-27 at a dose of 10 mg/kg combined with various doses of THC (0.3–3 mg/kg) did not change the THC-lever responses (Figure 8b).

## 3. Discussion

The current preliminary in vivo study aimed to evaluate the pharmacological effects of GPR18 activation and inhibition using novel selective/preferential ligands, including an agonist and two antagonists. The study sought to investigate the potential role of GPR18 in food intake, body temperature regulation, mood disorders, and acute and chronic (neuropathic) pain. We also examined whether GPR18 compounds share the discriminative stimulus properties with THC, which might reflect similarities between GPR18 and CB receptors in neurotransmission.

The first behavioral screen assessed the effects of the compounds on locomotor activity after a single intraperitoneal administration. The disruption of locomotor activity (both sedation and hyperactivity), which may cause false results in further tests, was also evaluated for toxicity signs (poor well-being) in animals [19,20]. We found that acute administration of the GPR18 agonist PSB-KK-1415 did not affect mice ambulation, indicating that receptor stimulation was not associated with changes in locomotor activity. However, acute administration of the GPR18 antagonists PSB-CB-5 or PSB-CB-27 significantly reduced locomotor activity at high doses of 30 mg/kg and 10 mg/kg, respectively, suggesting that the pharmacological blockade of the receptor may inhibit tonic receptor activation controlling locomotor activity. Another possible explanation for the reduced motility could be off-target effects. Therefore, these high doses of GPR18 antagonists were excluded from subsequent tests. It must be emphasized that the specific compounds used in the present study may have other biological targets that are currently unknown. Hence, the study is considered preliminary, and the precise sequence of molecular events in the organism that link the influence of the tested compounds on GPR18 with locomotor activity, depression-like and anxiety-like symptoms, pain threshold, core temperature, and food intake, remains an open question that requires further research.

Bearing in mind the anxiolytic, anti-stress, and antidepressant effects of endoCB signaling, we investigated whether GPR18 ligands can affect emotional responses. The rodent forced swim test is a primary screening test for antidepressants [21]. This test also provides a useful model to study neurobiological and genetic mechanisms underlying stress responses [22]. In our study, it was found that among the investigated GPR18 ligands, only the GPR18 agonist, PSB-KK-1415, significantly and specifically (lack of changes in locomotor activity) decreased the immobility time at two doses; however, a dose-dependent response was not observed for this behavioral outcome. Of note, a study investigating the antidepressant-like activity of THC also showed a U-shaped dose-response curve for THC [23]. THC is not only a partial CB_1_/CB_2_ receptor agonist, but also a moderately potent agonist of GPR18 [6]. Whether the antidepressant-like activity of PSB-KK-1415 and THC is dependent on GPR18 receptor stimulation requires further studies using selective receptor antagonists.

In the four-plate test, an animal test useful for studying anxiety [24], we demonstrated that the GPR18 agonist PSB-KK-1415 (significantly at a dose of 1 mg/kg and non-significantly by 62% at a dose of 10 mg/kg) increased the number of spontaneous punished crossings, which means that stimulation of GPR18 is associated with an anxiolytic-like effect, and GPR18 may even be involved in the anti-anxiety response of THC [25]. In fact, other studies indicate that pharmacological blockade of endocannabinoid metabolism elicits anxiolytic- and antidepressant-like effects in rodents [26]. Additional research is also required to explain the anxiolytic-like effect of a high dose of the preferential GPR18 receptor antagonist PSB-CB-5, keeping in mind that this compound has some affinity for the CB_2_ receptor [15], in contrast to the more selective GPR18 antagonist PSB-CB-27, which did not show activity in the four-plate test.

It should be underlined that the drugs’ activity in the four-plate test might be confounded by the antinociceptive properties of tested compounds. In other words, such drugs might give false positive results in the four-plate test. In the present study, we additionally studied the effects of GPR18 compounds on pain sensitivity in the hot plate test. We observed that neither the GPR18 agonist PSB-KK-1415 nor the antagonist PSB-CB-5 (at doses active in the four-plate test) affected the reaction time to the thermal stimulus in the hot plate test, which means that their effects observed in the four-plate test do reflect anxiolytic-like activity. Of note, a role of GPR18 in pain perception has been previously suggested [27]. Interestingly, in our study, a significant reduction in the reaction time to pain induced by a thermal stimulus, controlled by supraspinal mechanisms [28], was observed up to 60 min following administration of the lowest (0.3 mg/kg) dose of the selective GPR18 antagonist PSB-CB-27. This effect requires further research; however, it indicated a potential facilitation of nociceptive transmission induced by PSB-CB-27 administration (i.e., due to GPR18 antagonism).

In the next set of experiments, we investigated the effects of GPR18 ligands on symptoms of chronic neuropathic pain. For this purpose, we used a mouse model of chemotherapy-induced peripheral neuropathy (CIPN) caused by oxaliplatin. Oxaliplatin is a third-generation platinum-based antineoplastic agent, which alkylates DNA, thus inhibiting DNA synthesis [29]. Interestingly, CIPN induced by oxaliplatin manifests in two clinically distinct forms, i.e., an acute one and a chronic one, with painful symptoms (tactile allodynia and cold allodynia/hyperalgesia) triggered by mechanical stimuli and exposure to cold [30]. The mechanisms underlying neuropathic pain caused by oxaliplatin are not fully established [31,32]; however, the involvement of neuroinflammation [33] and/or CB_1_ receptor activation [34,35] in the development of this type of pain have been proposed. Here we assessed potential antiallodynic and antihyperalgesic activities of PSB-KK-1415 and PSB-CB-27 in oxaliplatin-treated mice. The agonist PSB-KK-1415 at the highest dose tested (30 mg/kg) attenuated tactile allodynia in oxaliplatin-treated neuropathic mice. At the same time, this dose was not effective in the attenuation of cold hyperalgesia. Taken together, these findings may indicate (and correspond well with literature data (e.g., [36])) that both tactile and cold hyperalgesia induced by oxaliplatin are mediated by distinct mechanisms, and that GPR18 stimulation might be involved in the attenuation of mechanical allodynia. Interestingly, PSB-KK-1415 reduced only late-phase tactile allodynia, while it was not able to attenuate early-phase tactile allodynia in oxaliplatin-treated mice. This also suggests that tactile allodynia at both time points develops via distinct mechanisms, and only the late-phase symptoms could be modulated by mechanisms related to GPR18 activation. We also observed that the GPR18 antagonist PSB-CB-27 at a dose of 30 mg/kg lowered the cold pain threshold in oxaliplatin-treated mice. Although the GPR18 agonist that we used in this study at a dose of 30 mg/kg did not affect cold hyperalgesia, the antagonist of this receptor reduced the latency to the pain reaction in the late phase of CIPN. Our finding mimics previous observations showing that GPR18 might be involved in pain sensation [27]. Of note, the action of PSB-CB-27 in the cold plate test resembles that observed for the compound at 0.3 mg/kg in the hot plate test and further supports a potential role of GPR18 in thermally-induced pain.

So far, the role of GPR18 in the regulation of body temperature has not yet been studied. Keeping in mind that pharmacological CB_1_/CB_2_ receptor stimulation with THC (but not tonic CB_1_ receptor activity following selective CB_1_ or CB_2_ receptor antagonists) reduces body temperature (the present paper and [37,38,39]), our study showed that, at any dose used, GPR18 antagonists did not significantly affect the core temperature of mice. These compounds were also unable to reverse THC-induced hypothermia. Therefore, GPR18 does not appear to play a crucial role in CB-induced hypothermia.

Next, the role of GPR18 in regulating eating behavior was investigated. Our research clearly showed that the agonist of GPR18, PSB-KK-1415 (at the highest dose), and the antagonist, PSB-CB-5 (at all doses used), decreased food intake in mice within 2 h of special feeding. Interestingly, the second selective antagonist of GPR18, PSB-CB-27, did not influence this behavior. Whether PSB-CB-5 functions as a partial agonist in this assay or its effects are linked to CB receptor blockade is unclear, and more research is needed to clarify these questions. These observations are not in line with the previously published results of influence on the food intake studies in the rat model of excessive eating, where we showed that PSB-KK-1415 increases caloric intake during three weeks of administration but both PSB-CB5 and PSB-CB27 reduced caloric intake [40]. We suggest that these discrepancies between the results may be due to the different effects of the tested compounds after a single and repeated administration. Of note, in animal models, the exogenous administration of CBs produces biphasic effects, i.e., hyperphagia is observed at low doses while hypophagia at high doses [41,42]. This may be also related to receptor up- and/or down-regulation after repeated compound administration, and will require further investigation.

Finally, we investigated whether our novel ligands of GPR18 could change the subjective effects of THC. In line with previous studies [43,44,45], THC served as an effective discriminant stimulus, resulting in a dose-dependent substitution of the training dose of THC. In further substitution studies, neither PSB-KK-1415 nor both GPR18 receptor antagonists PSB-CB-5 or PSB-CB-27 mimicked the subjective effects of THC. Of note, partial substitution is considered as >60% drug lever-responding [18]. Similarly, in combination tests, PSB-KK-1415, PSB-CB-5, or PSB-CB-27 co-administered with THC did not alter the dose-response curved for the training drug. Our findings suggest that GPR18 does not play a critical role in the subjective effects of THC. Previous data indicated that the discriminating effects of THC depend on stimulation of CB_1_ receptors as their selective agonists mimicked THC-stimulus properties, while selective CB_1_ receptor antagonists attenuated the effect [43,45]. This indicates that while THC exerts some agonistic activity toward GPR18, its subjective effects are mediated by CB_1_ receptors rather than GPR18.

Further investigations are necessary to fully elucidate the potential of GPR18 as a therapeutic target for mood, pain, and/or eating disorders.

## 4. Materials and Methods

### 4.1. Drugs, Chemical Reagents, and Other Materials

In our experiments, three ligands of GPR18 were used: an agonist (PSB-KK-1415 with an undisclosed structure) and two antagonists (PSB-CB-5-(Z)-2-(3-(4-chlorobenzyloxy)benzylidene)-6,7-dihydro-2*H*-imidazo[2,1-*b*][1,3]thiazin-3(5*H*)-one) and PSB-CB-27 ((Z)-2-(3-(6-(4-chlorophenoxy)hexyloxy)benzylidene)-6,7-dihydro-2*H*-imidazo[2,1-*b*][1,3]thiazin-3(5*H*)-one), discovered in the Department of Pharmaceutical and Medicinal Chemistry, University of Bonn, Germany, and synthesized in the Department of Technology and Biotechnology of Drugs, Jagiellonian University, Poland. The method of PSB-CB-5 synthesis was described by Rempel et al. [14], while that of PSB-CB-27 was characterized by Schoeder et al. [15]. The compounds were suspended in 1% Tween-80 (Sigma Aldrich, Schnelldorf, Germany) and administered as an intraperitoneal injection (i.p.) 30 or 45 min (drug combination) before behavioral tests. Δ^9^-Tetrahydrocannabinol hydrochloride (THC; THC Pharm GmbH, Illertissen, Germany) was prepared in a mixture of propylene glycol (Sigma Aldrich, Germany) and Tween-80 (1:1, *v*/*v*), and stored at −20 °C until used. The THC working solution was prepared before test sessions by dissolving the stock solution in sterile saline (0.9% NaCl). THC was administered as an i.p. injection 30 min before the tests.

Oxaliplatin was purchased from Activate Scientific GmbH (Germany), and prepared in a 5% glucose solution (Polfa Kutno, Kutno, Poland).

All compounds were used at a volume of 1 mL/kg for rats or 10 mL/kg for mice.

### 4.2. Animals

The present in vivo study was conducted in the Maj Institute of Pharmacology, Polish Academy of Sciences, Krakow, Poland (procedures described in Section 4.3, Section 4.4, Section 4.5, Section 4.6, Section 4.7, Section 4.9, and Section 4.10), and the Faculty of Pharmacy, Jagiellonian University Medical College, Krakow, Poland (the procedure described in Section 4.8).

All animals were delivered by licensed breeders (procedures 4.3–4.7, 4.9, and 4.10: Charles River Laboratories, Germany; procedure 4.8: Animal breeding farm of the Faculty of Pharmacy of the Jagiellonian University Medical College) and were housed in standard plastic rodent cages in a colony room maintained at 22 ± 2 °C and at 45–65% humidity under a 12-h light–dark cycle (lights on at 06:00 a.m.). Male Albino Swiss mice (20–25 g, 6 weeks old) had free access to food (Labofeed pellets VRF1, Essex, UK) and water during experiments, unless otherwise stated (4.7). Male Wistar Han rats (250–300 g, 8 weeks old) had free access to food (Labofeed pellets) and water during the 7-day habituation period. They were then maintained on limited water during drug-discrimination procedures (4.10). Each experimental group consisted of 6 to 13 animals. An observer blinded to the treatment did all behavioral measurements. All experiments were conducted during the light phase of the light–dark cycle (between 8.00 a.m. and 3.00 p.m.). The experiments were carried out in accordance with ethical standards laid down in respective Polish regulations and European Directive 2010/63/EU and were approved by the Local Ethical Committee at the Institute of Pharmacology, Polish Academy of Sciences (permission numbers: 30/2016, 40/2016, 219/2016).

### 4.3. Locomotor Activity

The locomotor activity was recorded individually for each animal in OPTO-M3 locomotor activity cages (Columbus Instruments, Cleveland, OH, USA) linked online to a compatible PC. Each cage (13 cm × 23 cm × 15 cm) was surrounded by an array of photocell beams. Interruptions of these photobeams resulted in horizontal activity defined as ambulation scores. Mice were placed separately into activity cages and the ambulation scores were measured for 60 and 120 min.

### 4.4. Forced Swim Test

The experiment was carried out according to the method of Porsolt et al. [21], with small modifications. Briefly, mice were individually placed in a glass cylinder (25 cm high, 10 cm in diameter) containing 11 cm of water maintained at 23–25 °C and were left therein for 6 min. A mouse was regarded as immobile when it remained floating on the water making only small movements to keep its head above it. The total duration of the immobility was measured during the final 4 min of a 6-min test session.

### 4.5. Four-Plate Test

The four-plate apparatus (Bioseb, Vitrolles, France) consists of a cage (25 cm × 18 cm × 16 cm) that is floored with four rectangular metal plates (11 cm × 8 cm). The plates are separated from one another by a gap of 4 mm, and they are connected to an electroshock generator. The test was performed according to Bourin et al. [46]. After the habituation period (15 s), each mouse was subjected to an electric shock (0.8 mA, 0.5 s) when crossing from one plate to another (two limbs on one plate and two on another). The number of punished crossings was counted during 60 s.

### 4.6. Hot Plate Test

The hot plate apparatus (Hot/Cold Plate, Bioseb, Vitrolles, France) consists of an electrically heated surface, and it is equipped with a temperature controller that keeps the temperature constant at 55–56 °C. The test was performed as previously described [47]. Before the experiment, the animals were tested for their pain sensitivity threshold (baseline latency). For further pain tests, only mice with baseline latencies ≤ 30 s were selected. The latency to pain reaction (licking hind paws or jumping) was measured as the indicator of nociception. The cut-off time was established (60 s) and animals that did not respond within 60 s were removed from the hot plate apparatus and assigned a score of 60 s.

### 4.7. Food Intake

The mice underwent a 24-h food deprivation period before the testing phase, during which animals were provided with free access to water. On the test day, mice were given either drugs or a vehicle. Subsequently, pre-weighed food pellets (ca. 3 g) were placed centrally in a novel test environment (an autoclaved transparent glass container, 16 cm in height and 9.5 cm in diameter). The mice were then placed individually in the containers, each covered with a metal perforated lid enabling unrestricted respiration. After 2 h, the animals were returned to their home cages, with food and water available ad libitum. After 24 h, dried food remains were weighed again. The relative food intake was calculated as g of food consumed per kg of body weight per 2 h.

### 4.8. Oxaliplatin-Induced Neuropathic Pain

#### 4.8.1. Induction of Oxaliplatin-Induced Peripheral Neuropathy

To induce peripheral neuropathy, oxaliplatin was administered as a single i.p. dose of 10 mg/kg 3 h before behavioral tests.

#### 4.8.2. Assessment of Mechanical Nociceptive Threshold—Von Frey Test

Trained observers blinded to experimental conditions scored the behavioral measures. Mechanical hypersensitivity (tactile allodynia) was assessed using the electronic von Frey unit (Bioseb, Vitrolles, France) supplied with a single flexible filament applying increasing force (from 0 to 10 g) against the plantar surface of the hind paw of the mouse. The nocifensive paw withdrawal response automatically turned off the stimulus and the mechanical pressure that evoked the response was recorded.

On the day of the experiment, the mice were placed individually in test compartments with a wire mesh bottom and were allowed to habituate for 1 h. After the habituation period, in order to obtain baseline (pre-drug) values of pain sensitivity, each mouse was tested 3 times alternately in each hind paw, allowing at least 30 s between each measurement. Then, the mice were pretreated with the test compound or vehicle. Sixty min later, the animals were tested again and the mean post-drug values of the paw withdrawal threshold were obtained for each mouse. In oxaliplatin-treated mice, the baseline paw withdrawal threshold was measured 3 h (assessment of early-phase allodynia) and 7 days (assessment of late-phase allodynia) after oxaliplatin injection [48].

#### 4.8.3. Assessment of Thermal (Cold) Nociceptive Threshold—Cold Plate Test

The cold plate test was performed using the hot/cold plate apparatus (Bioseb, Vitrolles, France) set at 2.5 °C. In this assay, the animals were tested first to obtain baseline latencies to the pain reaction (i.e., lifting, biting, shaking of hind paws, jumping, and movement deficits) before oxaliplatin injection (referred to as latencies ‘before oxaliplatin’). Then, oxaliplatin was injected and 3 h later, the latencies to pain reaction were measured again (referred to as ‘pre-drug’ latencies). Next, test compounds were administered and ‘post-drug’ latencies were measured. In this assay, a cut-off time of 60 s was established to avoid potential paw tissue damage and animals not responding within 60 s were removed from the apparatus and assigned a score of 60 s [49].

### 4.9. Body Temperature

Effects of the THC, PSB-CB-5, and PSB-CB-27 given alone or in combination on the rectal body temperature in mice (measured with an Ellab thermometer, Copenhagen, Denmark) were recorded 30 min after the last drug administration for 120 min, as described previously [50]. The basal body temperature was measured twice at the beginning of the experiment (−90 and −60 min).

### 4.10. Drug Discrimination

The procedure was performed according to the previous studies [43,45]. Rats had restricted access to water during the daily training sessions (5–6 mL/rat/session), after test sessions (15 min), and free access to water from Friday till Sunday afternoon. Rats were trained to discriminate THC (3 mg/kg) from sodium chloride (i.p., NaCl 0.9%) in sessions that lasted 15 min. Briefly, THC or saline was administered 30 min before daily (Monday–Friday) sessions in two-lever standard operant chambers (Med-Associates, Latham, NY, USA) under a fixed-ratio 10 schedule of reinforcement (FR 10) of continuous water reinforcement and depending on the treatment, left or right lever became active. That phase of training continued until the animals met the criterion (an individual mean accuracy of at least 80% of correct responses, before the first reinforcer during 10 consecutive sessions). During this phase, 3 rats were excluded from the study for failure to maintain performance at the criterion level (above). Later, test sessions were conducted once or twice a week, while THC and saline sessions intervened between the test sessions to maintain discrimination accuracy. Only the rats that met an 80% performance criterion during the preceding THC and saline sessions were used in the tests. After the completion of 10 responses to either lever or after the session time elapsed, a single reinforce was delivered and the animals were removed from the chamber. In substitution tests, rats were tested with different doses of THC (0–3 mg/kg; 30 min before the test), PSB-KK-1415 (1–10 mg/kg, 5 min before the test and measured by 90 min), PSB-CB-5 (10 mg/kg) and PSB-CB-27 (10 mg/kg) 30 min before the test. In combination tests with THC (0.3–3 mg/kg), PSB-CB-5 (10 mg/kg) and PSB-CB-27 (10 mg/kg) were given 45 min before the test, and PSB-KK-1415 (10 mg/kg) were given 35 min before the test.

In rats, during training sessions, accuracy was defined as the percentage of correct responses to total responses before the delivery of the first reinforcer; during test sessions, performance was expressed as the percentage of THC-appropriate responses to total responses before the delivery of the first reinforcer. Response rates (responses per minute) were also evaluated during training and test sessions as a measure of behavioral disruption. For training sessions, the response rate was calculated as the total number of responses emitted on either lever before completion of the first FR 10 divided by the number of minutes taken to complete that FR 10. During test sessions, the response rate was calculated as the total number of responses before the completion of 10 responses on either lever divided by the number of minutes necessary to complete the FR 10. Only the data from animals that completed the FR 10 during test sessions were used. A drug was considered to fully substitute for THC if at least 80–100% of responses occurred on the drug-appropriate lever after a dose of that drug; while complete antagonism was claimed to occur when no more than 20% drug-lever responding occurred after pretreatment with a dose of a potential antagonist given in combination with THC (0.3–3 mg/kg).

### 4.11. Data Analysis and Statistical Procedures

Data were analyzed using GraphPad Prism Software (v.5 or v.8, San Diego, CA, USA) or TIBCO Statistica v.13.3 (TIBCO Software, Palo Alto, CA, USA). For each parameter, the normality of the distribution was assessed using the Shapiro–Wilk test, while equality of variance was examined by the Levene test. After checking all the assumptions (i.e., normal distribution, equality of variance), the appropriate statistical tests were applied. The one or two-way repeated measures ANOVA was followed by post hoc Tukey’s or Bonferroni’s multiple comparison tests. For cases when assumptions for parametric tests were not met, the Kruskal–Wallis H or Friedman test with post hoc Dunn’s or multiple comparisons (2-tailed) tests was applied. Log-probit analyses were used to estimate the dose of THC predicted to elicit 50% drug-appropriate responding (ED_50_) and 95% confidence limits (CL) for each treatment combination [51]. Data dedicated to the effects of PSB-KK-1415, PSB-CB-5, PSB-CB-27, and THC in mice are expressed as the means and standard deviation of the mean (±SD) and in rats as mean or mean percentage and standard error of the mean (±SEM). The alpha value was set at 0.05.

## 5. Conclusions

In conclusion, our screening analyses to evaluate the pharmacological effects of novel GPR18 ligands indicated the impact on locomotor activity, depression- and anxiety-like symptoms, and pain threshold, and are similar to those of CB receptor activation. The complexity of orphan GPR18 receptors is important to consider in the subsequent development of therapeutic applications as a novel therapeutic target for mood and pain disorders. Further investigation is warranted to discern its function better.

## Figures and Tables

**Figure 1 ijms-24-09046-f001:**
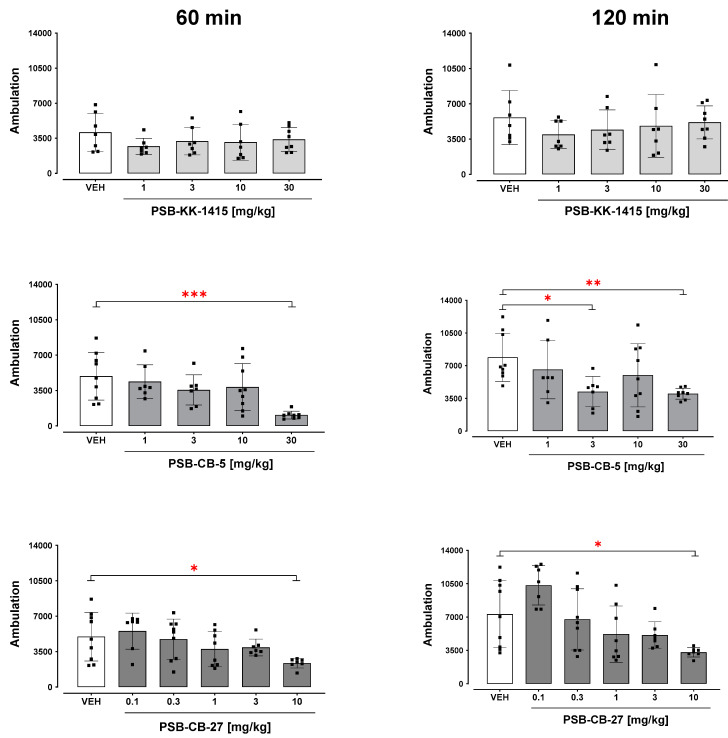
Locomotor activity after administration of PSB-KK-1415 (1–30 mg/kg), PSB-CB-5 (1–30 mg/kg), and PSB-CB-27 (0.1–10 mg/kg) in mice. The results for 60 (left panels) and 120 min (right panels) of measurement are expressed as the means ± SD of the data from 7–9 mice/group. * *p* < 0.05, ** *p* < 0.01, *** *p* < 0.001 versus vehicle (VEH).

**Figure 2 ijms-24-09046-f002:**
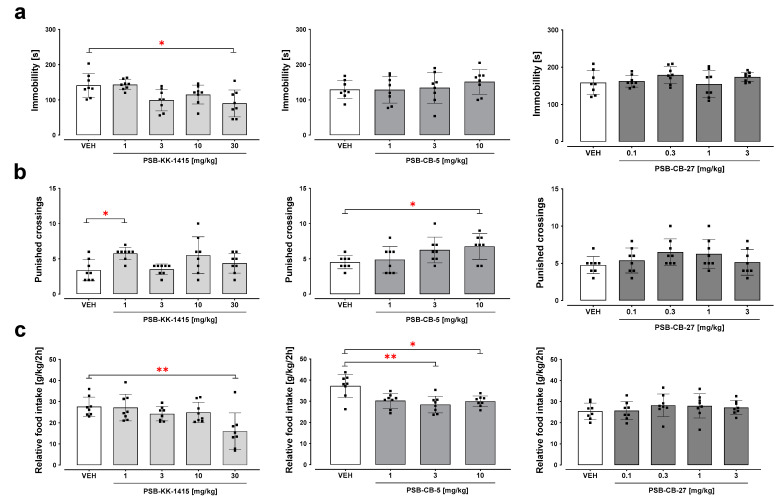
Forced swim test (**a**), four-plate test (**b**), and food intake (**c**) after administration of PSB-KK-1415 (1–30 mg/kg), PSB-CB-5 (1–10 mg/kg), and PSB-CB-27 (0.1–3 mg/kg) in mice. Relative food intake (g of food consumed per kg body weight per 2 h; g/kg/2 h). The results are expressed as the means ± SD from 8 mice/group. * *p* < 0.05, ** *p* < 0.01 versus vehicle (VEH).

**Figure 3 ijms-24-09046-f003:**
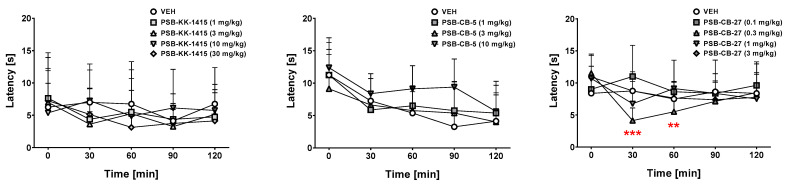
Effects of PSB-KK-1415 (1–30 mg/kg), PSB-CB-5 (1–10 mg/kg), and PSB-CB-27 (0.1–3 mg/kg) on the latency time in the hot plate test in mice. The results are expressed as the means ± SD of the data from 8 mice/group. ** *p* < 0.01, *** *p* < 0.001 versus the corresponding group at time 0. VEH—vehicle.

**Figure 4 ijms-24-09046-f004:**
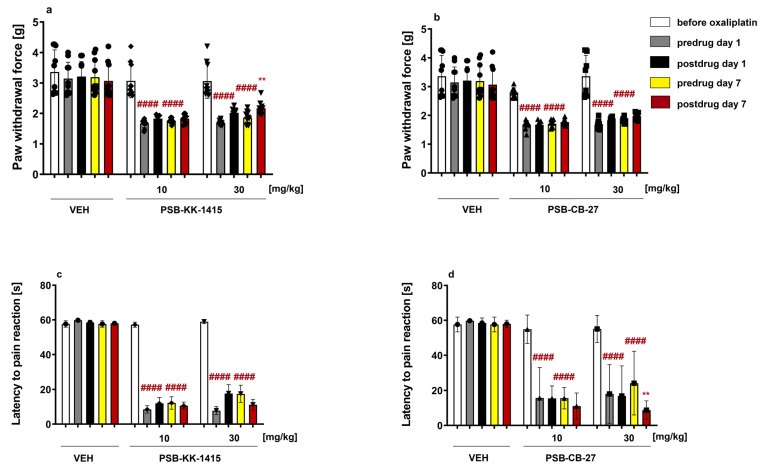
Effect of PSB-KK-1415 and PSB-CB-27 on tactile allodynia measured in the von Frey test (**a**,**b**) and cold hyperalgesia assessed using the cold plate test (**c**,**d**) in the oxaliplatin-induced neuropathic pain model in mice. Results are shown as the mean force that caused paw withdrawal (**a**,**b**), or the latency to pain reaction (**c**,**d**) ± SD for 8–10 mice/group. Statistical analysis: two-way repeated measures ANOVA followed by Bonferroni’s multiple comparison. Significance versus paw withdrawal force, or latency before oxaliplatin administration: ^####^
*p* < 0.0001; significance versus pre-drug values in the individual group: ** *p* < 0.01. VEH—vehicle.

**Figure 5 ijms-24-09046-f005:**
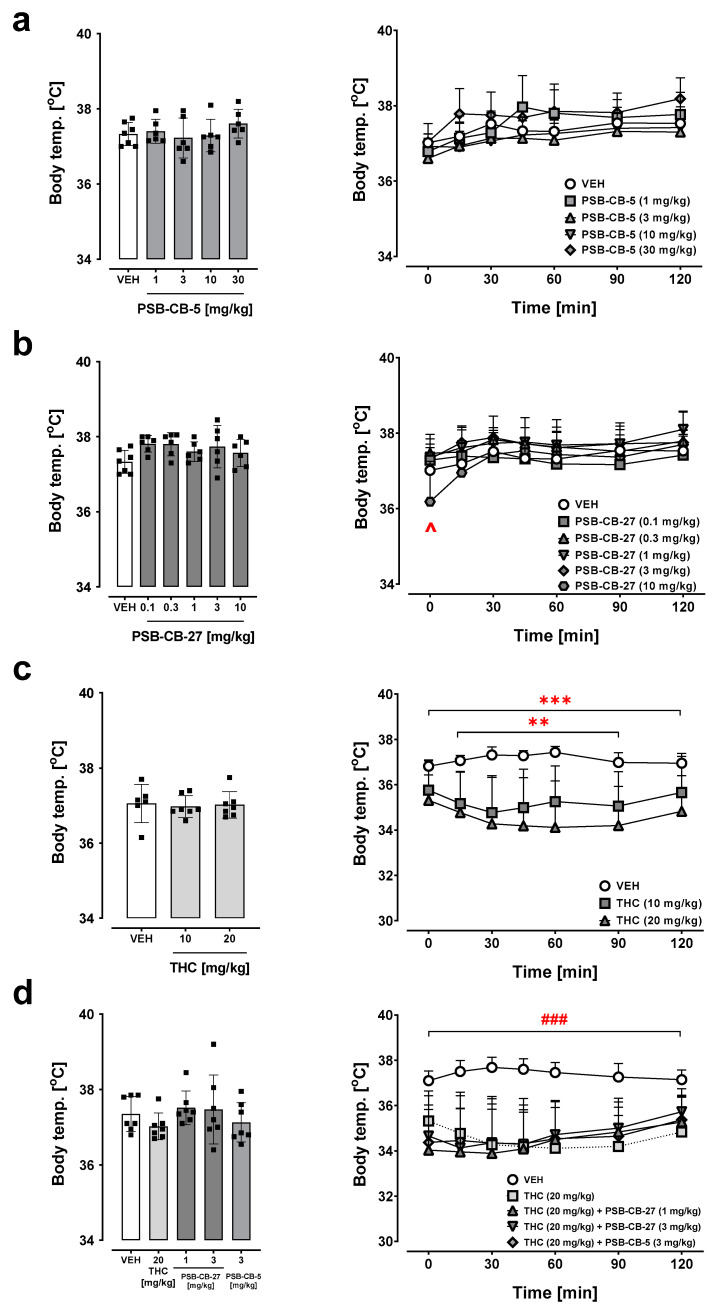
Effects of PSB-CB-5 (1–30 mg/kg) (**a**), PSB-CB-27 (0.1–10 mg/kg) (**b**), THC (10–20 mg/kg) (**c**), and their co-administration (**d**) on rectal body temperature in mice. The results are expressed as the means ± SD of the data from 6–7 mice/group; ^ *p* < 0.05 PSB-CB-27 10 versus vehicle (VEH); ** *p* < 0.01 THC 10 and *** *p* < 0.001 THC 20 versus VEH, ^###^
*p* < 0.001 THC 20 + PBS-CB-5 or THC 20 + PBS-CB-27 versus VEH.

**Figure 6 ijms-24-09046-f006:**
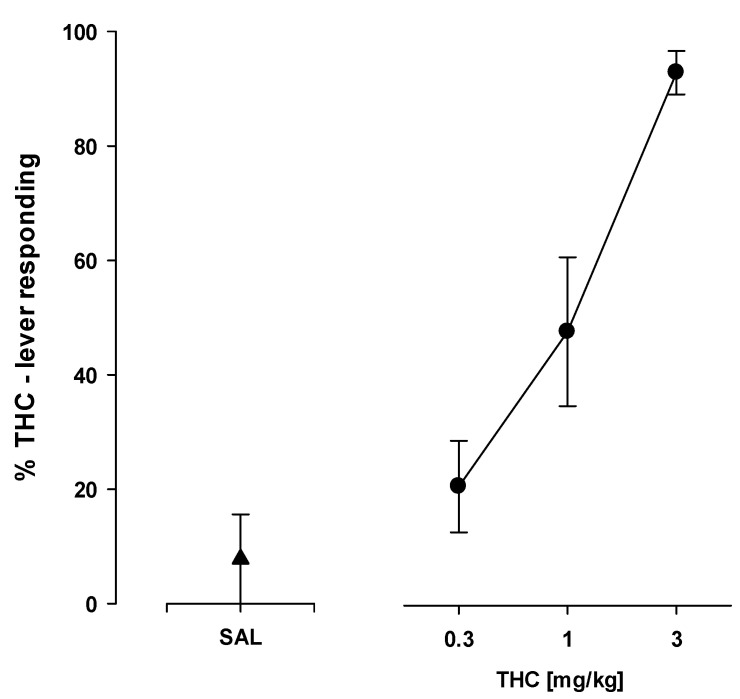
Drug-response effects in rats trained to discriminate THC (3 mg/kg) from saline (SAL). Symbols show the mean percentage of THC-lever responses ± SEM (closed symbols) and the mean number of responses/min ± SEM (open symbols). Performance is shown after injection of SAL (1 mL/kg; triangle) or THC (0.3–3 mg/kg; circles). All data points represent the means of data from 13/13 rats [n/N, number of rats (n) completing the fixed-ratio 10 schedule of reinforcement (FR 10) on either lever out of the number of rats tested (N)].

**Figure 7 ijms-24-09046-f007:**
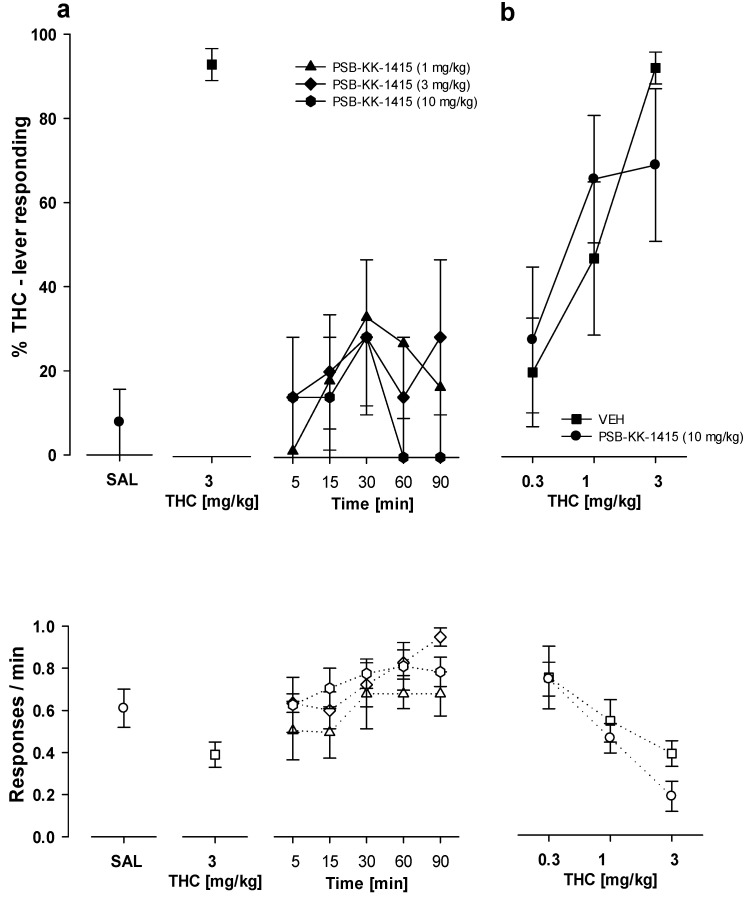
Substitution and combination studies with PSB-KK-1415 in rats trained to discriminate THC (3 mg/kg) from saline (SAL). Performance is shown after injection of PSB-KK-1415 (1, 3, 10 mg/kg) alone (**a**) or PSB-KK-1415 (10 mg/kg) in combination with THC (0.3–3 mg/kg) (**b**). Symbols show the mean percentage of THC-lever responses ± SEM (closed symbols) and the mean number of responses ± SEM (open symbols). All the data points represent the means of data from 5–8 rats.

**Figure 8 ijms-24-09046-f008:**
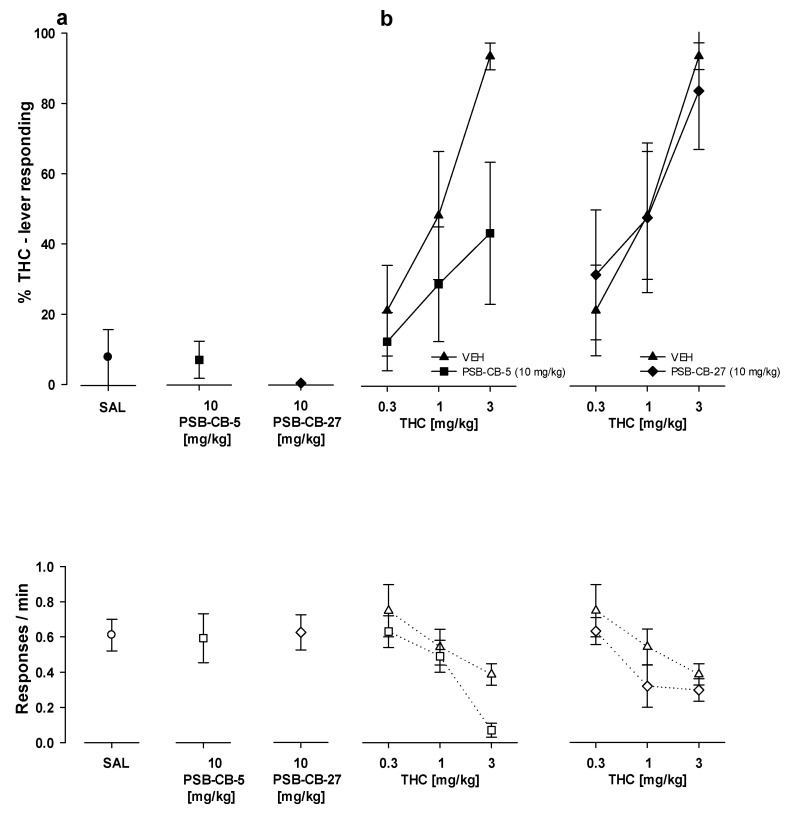
Substitution and combination studies with PSB-CB-5 and PSB-CB-27 in rats trained to discriminate THC (3 mg/kg) from saline (SAL). Performance is shown after injection of PSB-CB-5 (10 mg/kg) and PSB-CB-27 (10 mg/kg) alone (**a**), or PSB-CB-5 (10 mg/kg) and PSB-CB-27 (10 mg/kg) in combination with THC (0.3–3 mg/kg); (**b**). Symbols show the mean percentage of THC lever responses ± SEM (closed symbols) and the mean number of responses/min ± SEM (open symbols). All the data points represent the means of data from 6–8 rats.

## Data Availability

Data are available on request from the authors of individual results or corresponding authors.

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
