# Peer review of "Novel GPR18 Ligands in Rodent Pharmacological Tests: Effects on Mood, Pain, and Eating Disorders"

_ijms, 2023, doi:10.3390/ijms24109046_

Round 1

Reviewer 1 Report (Previous Reviewer 1)

The authors did not revise the manuscript and their response, compared to last version.

Author Response

Reviewer 2 Report (New Reviewer)

Summary:The authors investigated the pharmacological effects of the novel GPR18 ligands. They believe that these tools will help to investigate the role of orphan receptor GPR18.In conclusion, they mentioned that the GPR18 may have a role in mood, pain and/or eating disorders.

Conclusions:

1.In the abstract please modify the sentence in the line number 23 and 25.

2.The reviewer thinks it would be better if the authors could include a small introduction about the GPCRs before diving into introducing GPR18.

3.Among gastrointestinal, immune and testicular tissues, sperm, 44 and metastatic melanoma as well as various brain structures, where was GPR18 most abundantly expressed.

4.Upon brief PubMed search, the reviewer found another GPR18 agonist called PSB-KD-107used in some of the studies. The reviewer is curious to know if they have sued this agonist in their studies.

5.What was the age of the mice used in this study? Could the authors include this in the methods section?

Author Response

Reviewer 3 Report (New Reviewer)

The present study is solid and looks really positive to the field, but I am curious that if the authors considered doing in-silico study? If not, why? Is there any done research indicated that the study objects have such potential or does your idea come from elsewhere (literature-based as present in the current text only)? 

Author Response

Reviewer 4 Report (New Reviewer)

This article accounts the Novel GPR18 ligands in rodent pharmacological tests: effects on mood, pain, and eating disorders. Overall, it is well-written and explained in the manuscript. Following are few minor comments;

Comments 1. I will suggest authors to check more recent work/literature related to GPR18 ligands and their application. Some of the papers are still missing.

Author may be interested in: Front. Pharmacol. 9:1496, DOI:10.3389/fphar.2018.01496.

Comment 2. Check the correctness of the list of references, there are inaccuracies. Do it in the approved uniform MDPI style.

Comment 3. Specify how these compounds were selected, PSB-KK-1415, PSB-CB-5, and PSB-CB-27? Is there a synthetic part of the work. Describe the selection process in detail.

Comment 4. In Conclusion. "Our screening analyses to evaluate the pharmacological effects of novel GPR18 ligands indicated that the activity of this orphan G protein-coupled receptor is partly similar to that of the CB receptors with respect to their impact on emotional behaviors, food intake, and pain sensation." This sentence should be rewritten. Selected examples of the use of GPR 18 ligands should be disclosed carefully in conclusion.

Author Response

Reviewer 5 Report (New Reviewer)

In this study, the authors have studied effects on mood, pain, and eating disorders with respect to  GPR18 ligands in rodent. The manuscript is well-written and experimental work look appropriate.  The author should check for grammatical errors throughout the manuscript. References in text are mentioned with author name and years whereas in reference section they are mentioned with numbers. The whole reference section should be in a uniform pattern as per the journal format. 

Author Response

Reviewer 6 Report (New Reviewer)

I have doubts about whether this manuscript suits the International Journal of Molecular Sciences. It concerns a defined molecule, the G protein-coupled GPR1 receptor but the methods of assessing its function are behavioral. This approach is fully justified but far from studying molecular interactions. Moreover, the variability of results effects, quite understandable in animal behavioral studies makes the results not very convincing for a researcher used to molecular and cellular effects. It would be more suited for a pharmacological journal whose readers are used to such type of data.

As always, there is a question of specificity of the antagonists and agonist of the receptor. This point should be discussed since the lack of absolute specificity cannot exclude the mediation of the effects by other receptors.

Nevertheless, I find the manuscript interesting and bringing some (though not too much) new knowledge on the functions of the GPR1 receptor.

Figures: The symbols indicating statistical significance do not differ too much from symbols representing individual experimental points unless the Figure is magnified. Perhaps the use of color for the asterisk would help.

Please describe the axis of abscissae. The meaning of numbers “1”, “3” etc. is not obvious; I guess these are doses in mg/kg bm but it should be unequivocally stated.

Author Response

This manuscript is a resubmission of an earlier submission. The following is a list of the peer review reports and author responses from that submission.

Round 1

Reviewer 1 Report

Three GPR18 ligands, one agonist (PSB-KK-1415) and two antagonists (PSB-CB-5 and PSB-CB-22 27) were tested in male mice or rat to study the relationship between GPR18 and the cannabinoid (CB) receptor system, and the control of 24 endoCB signaling over sleep, emotions, food intake, pain sensation, and thermoregulation. The authors measured locomotor activity, depression-like and anxiety-like symptoms, pain thresholds, core temperature, food intake and THC-vehicle discrimination in multiple different experimental set up.

I don't suggest this type of experimental design. Eight different experiments were used to study the impacts of the drugs. Only animal behaviors are measured. No deep analysis or quantification for the  GPR18 or the signaling that the authors tried to claim was performed. No results are conclusive. The author should be more focused on one or two experiments and dig into the underlying mechanism.  

Reviewer 2 Report

01

No power analysis was performed. Therefore, it is not possible to know whether the analysis of the results of the present study is a true finding or a pure chance. This may compromise the entire validity of this study.

02

There was a very limited number of samples per group. Yet, the authors opted automatically for parametric tests (ANOVA and Student’s t-test).

03

“Further investigations will be required to fully elucidate the therapeutic potential of GPR18 as a therapeutic target for mood, pain, and/or eating disorders.”

Is this not more a recommendation than a conclusion? Should it not stay at the end of the discussion?

04

There are some sentences in the text without reference to a previous study (or studies) in order to give evidence to their statements. Without references, these statements would be mere assumptions or allegations by the authors of the manuscript. Therefore, each of the following sentences need at least one reference to back up their statement:

“The disruption of locomotor activity (both sedation and hyperactivity) may cause false results in the further tests or it may partly be associated with signs of toxicity (poor well-being of the animals).”

“Another explanation could be that the observed reduced motility is due to off-target effects.”

“THC is not only a partial CB 1 /CB 2 receptor agonist, but also a moderately potent agonist of GPR18.”

Round 2

Reviewer 1 Report

The authors discussed about 'the relationship between GPR18 and the cannabinoid (CB) receptor system, and the control of endoCB signaling over sleep'. This has to be shown in experiments as mechanism.

Reviewer 2 Report

Concerning my comments regarding the statistics:

·         About ANOVA: the authors’ response show that they lack basic knowledge on statistics.

·         About Student’s t-test: the authors misunderstood my comments. I am not talking about comparison of either two or three independent groups, but about the fact that the two groups being compared have small number of samples.

If the number of samples per group is low, the analysis calls for a non-parametric test regardless of the normality, as the normal distribution cannot properly be verified. Moreover, even if normality can be reasonably assumed, in small samples tests that assume normally distributed data are likely to be underpowered to detect departures from the equal variance assumption. That is, use of these tests in small samples may lead researchers to incorrectly conclude that the equal variance assumption is justified.

For more information the authors can consult, for example, the following literature (which is not limited by these): (a) Altman DG, Gore SM, Gardner MJ, Pocock SJ. Statistical guidelines for contributors to medical journals. Br Med J (Clin Res Ed). 1983;286(6376): 1489–1493; (b) Fagerland MW. t-tests, non-parametric tests, and large studies--a paradox of statistical practice?. BMC Med Res Methodol. 2012;12: 78; (c) Morgan CJ. Use of proper statistical techniques for research studies with small samples. Am J Physiol Lung Cell Mol Physiol. 2017;313(5): L873–L877; (d) Dwivedi AK, Mallawaarachchi I, Alvarado LA. Analysis of small sample size studies using nonparametric bootstrap test with pooled resampling method. Stat Med. 2017;36(14): 2187–2205; (e) Wilcox R, Peterson TJ, McNitt-Gray JL. Data Analyses When Sample Sizes Are Small: Modern Advances for Dealing With Outliers, Skewed Distributions, and Heteroscedasticity. J Appl Biomech. 2018;34(4): 258–261.

Therefore, the authors have to use Kruskal-Wallis test instead of ANOVA with the appropriate post-hoc test (I will not tell the authors which post hoc test they should use in this case. Let’s see if the authors will at least make the right decision here), and Mann-Whitney test instead of Student’s t-test.

Round 3

Reviewer 1 Report

My concerns are not solved in the revision.

Reviewer 2 Report

Concerning the use of ANOVA, the authors replied:

“However, to keep the presented results consistent with the other experiments described in the paper, we have decided to keep the results of the one-way ANOVA analyses and show the data as means ± SEM.”

This is, not at all, a reason to keep ANOVA. What the authors are doing is throwing the assumptions of a specific statistical test in the garbage bin, so “to keep the presented results consistent with the other experiments described in the paper”. Since when is this a scientific evidence-based justification?

An important component of evaluating reported experimental results is the ability to recognize misapplication of data analysis that can potentially result in misleading interpretation of results and a subsequent failure to replicate similar findings.

Though it is true that reporting, formatting, or display errors do not change the outcome of study results, misleading and inaccurate reporting can greatly influence the perception of the results, exaggerate the true merits of the study, lead investigators in the wrong direction, and bias future research based upon it, all contributing to the problems of non-replication. One example is the current predominance of reporting small sample sizes with the population statistic (appropriate for N>20) of standard error of the mean, rather than the proper sample measure, the SD. This practice makes the results of the outcome measure appear to have greater consistency within and between samples, which is not present, leading to overemphasized results from studies with small sample sizes and increasing the chances of results that are difficult to replicate.